# Polymorphism Detection of *GDF9* Gene and Its Association with Litter Size in Luzhong Mutton Sheep (*Ovis aries*)

**DOI:** 10.3390/ani11020571

**Published:** 2021-02-22

**Authors:** Fengyan Wang, Mingxing Chu, Linxiang Pan, Xiangyu Wang, Xiaoyun He, Rensen Zhang, Lin Tao, Yongfu La, Lin Ma, Ran Di

**Affiliations:** 1Key Laboratory of Animal Genetics, Breeding and Reproduction of Ministry of Agriculture and Rural Affairs, Institute of Animal Science, Chinese Academy of Agricultural Sciences, Beijing 100193, China; wangfy504@163.com (F.W.); mxchu@263.net (M.C.); xiangyu_wiggle@163.com (X.W.); hedayun@sina.cn (X.H.); rensenz@163.com (R.Z.); taolincanjian@163.com (L.T.); layongfu@yeah.net (Y.L.); 18633082661@163.com (L.M.); 2Ji’nan Laiwu Yingtai Agriculture and Animal Husbandry Technology Co., Ltd., Ji’nan 271114, China; yingtai2020@163.com

**Keywords:** sheep, *GDF9* gene, *FecB* gene, litter size

## Abstract

**Simple Summary:**

*GDF9* and *BMPR1B* are two important reproduction genes. In this study, the whole coding region of *GDF9* was sequenced, of which the mutations were detected in Luzhong mutton sheep. The results suggested that two single nucleotide polymorphisms (SNPs), g.41768501A > G and g.41768485 G > A in *GDF9* gene were associated with litter size. The g.41768485 G > A is a missense mutation which is predicted to affect the tertiary structure of the protein. Thus, these two mutations may be potential effective genetic markers to improve the litter size in sheep.

**Abstract:**

Litter size is one of the most important economic traits in sheep. *GDF9* and *BMPR1B* are major genes affecting the litter size of sheep. In this study, the whole coding region of *GDF9* was sequenced and all the SNPs (single nucleotide polymorphisms) were determined in Luzhong mutton ewes. The *FecB* mutation was genotyped using the Sequenom MassARRAY^®^SNP assay technology. Then, the association analyses between polymorphic loci of *GDF9* gene, *FecB,* and litter size were performed using a general linear model procedure. The results showed that eight SNPs were detected in *GDF9* of Luzhong mutton sheep, including one novel mutation (g.41769606 T > G). The g.41768501A > G, g.41768485 G > A in *GDF9* and *FecB* were significantly associated with litter size in Luzhong mutton ewes. The g.41768485 G > A is a missense mutation in the mature GDF9 protein region and is predicted to affect the tertiary structure of the protein. The results preliminarily demonstrated that *GDF9* was a major gene affecting the fecundity of Luzhong mutton sheep and the two loci g.41768501A > G and g.41768485 G > A may be potential genetic markers for improving litter size.

## 1. Introduction

Litter size is one of the most important economic traits and is closely related to the economic benefits of modern sheep farming. Study on the key genes of litter size can provide useful information for marker-assisted breeding to increase litter size.

The growth differentiation factor 9 (*GDF9*) gene, encoding one member of the transforming growth factor β (TGF-β) superfamily, has important functions in the reproduction of ewes [1]. The gene was mapped on sheep chromosome 5 [2] and contains two exons separated by an intron. *GDF9* is mainly expressed in oocytes and plays an important role in follicular development and ovulation in sheep [3]. In previous studies, many mutations in the gene were identified in Belclare sheep (*FecG^F^*) [4] of Europe, Iranian Afshari sheep (G2, G3, and G4) [5] and Fat-Tailed sheep (G1, G2, G3, and G4) [6] of Middle Eastern origin, and Rahmani sheep (*FecG^H^*) [7] of Africa. It was reported that c.978A>G and c.994G>A, *FecG^H^*, c.1040T>C (Phe 347 Ser) were associated with litter size in Araucana creole, Rahmani, and Mongolia sheep [7,8,9], respectively.

*BMPR1B* was the first identified major gene for the fertility of sheep [10] and was mapped to chromosome 6 of sheep [11]. BMPR1B protein is also one of the members of TGF-β superfamily [12]. *FecB* is an A746G mutation in *BMPR1B* coding sequence which causes nonconservative substitution (Q249R), and this mutation is related to the prolific phenotype in Australian (Booroola sheep) and Chinese (Small Tailed Han sheep) breeds [13,14,15,16].

Luzhong mutton sheep is a new breed with high fertility and meat performance. It was developed by crossbreeding White-headed Dorper sheep from South Africa and Hu sheep from China. These two progenitor breeds have medium or high fecundity. Therefore, Luzhong mutton sheep is likely to carry *FecB* and mutations in the *GDF9* gene simultaneously. Excavating mutations associated with litter size may help to discover new molecular markers to increase the efficiency of sheep reproduction. Thus, the objective in present study was to detect mutations in the *GDF9* gene and determine the mutation loci associated with litter size in Luzhong mutton sheep.

## 2. Materials and Methods

### 2.1. Animals and DNA Extraction

Jugular blood samples of 154 Luzhong Mutton ewes with litter size records of two parities were collected from Ji’nan Laiwu Yingtai Agriculture and Animal Husbandry Technology Co. Ltd. (Ji’nan, Shandong, China). Genomic DNA of these ewes were extracted by the phenol–chloroform method and then dissolved in ddH_2_O. All experimental procedures involved in this study were approved by the Animal Welfare Division of the Institute of Animal Science, Chinese Academy of Agricultural Sciences (IAS-CAAS) (Beijing, China). In addition, ethics approval was given by the animal ethics committee of IAS-CAAS (No. IAS2020-64) on 27 April 2020.

### 2.2. Full-Length Sequencing and Mutation Detection of GDF9 Coding Sequence 

To amplify the *GDF9* gene coding sequence in Luzhong mutton sheep, three pairs of primers were designed with Primer Premier software (version 5.0, PREMIER Biosoft international, San Francisco, CA, USA) software according to ovine *GDF9* sequence (GenBank NC_040256.1). Detailed information of the primers is shown in Table 1. PCR products were sequenced using the Sanger sequencing method in Suzhou Genewiz Biotechnique Co. Ltd. (Suzhou, China) with the primer the same as the amplification primer. After sequencing the PCR products, the entire *GDF9* coding sequence was assembled. Then, *GDF9* gene sequences of 154 Luzhong Mutton sheep were aligned with the reference sequence of the ovine *GDF9* gene (GenBank NC_040256.1) using MEGA7 software [17]. Finally, polymorphic loci of this gene were determined in Luzhong Mutton sheep and were subsequently searched in the Ensembl database (https://asia.ensembl.org/Ovis_aries/Gene/Variation_Gene/Table?db=core;g=ENSOARG00000013229;r=5:41841034-41843517;t=ENSOART00000014382, accessed on 27 May 2020) to check if these loci were novel mutations.

### 2.3. Genotyping of FecB Mutation

The *FecB* mutation was genotyped in Luzhong mutton ewes using the Sequenom MassARRAY^®^SNP assay method previously described by Zhang et al. [18].

### 2.4. Statistical Analysis

Allele and genotype frequency, polymorphism information content (*PIC*), heterozygosity (*He*), and number of effective alleles (*Ne*) were calculated for *GDF9* and *BMPR1B* genes using the following formula [19]: (1)$$He=1-\overset{n}{\underset{i=1}{\sum }}p_{i}{}^{2}$$
(2)$$Ne=1/\overset{n}{\underset{i=1}{\sum }}p_{i}{}^{2}$$
(3)$$PIC=1-\overset{n}{\underset{i=1}{\sum }}p_{i}{}^{2}-\overset{n-1}{\underset{i=1}{\sum }}\overset{n}{\underset{j=i+1}{\sum }}2p_{i}{}^{2}p_{j}{}^{2}$$
where *n* is the number of alleles, *p_i_* is the allele frequency of the *i*th allele, and *p_j_* is the allele frequency of the *j*th allele.

The genotype distribution of each locus was tested for deviation from the Hardy–Weinberg equilibrium using the Chi-Square test [20]. The association of litter size with the genotypes of *GDF9* and *FecB* was analyzed using the following fixed effects model in Luzhong mutton sheep, with least squares means used for multiple comparisons in litter size among the different genotypes: *y* =*μ* + *P* + *G1* + *G2* + *G1G2* + *e*, where *y* is the phenotypic value of litter size, *μ* is the population mean, P is the fixed parity effect (two levels), *G1* is the fixed effect for *GDF9* genotypes, *G2* is the fixed effect for *FecB* genotype, *G1G2* is the fixed interaction effect for *GDF9* and *FecB* combined genotypes, and *e* is the random error effect of each observation. Analysis was performed using the general linear model procedure by R software (aov, Version 4.0.3) [21]. Mean separation procedures were performed using the Duncan test, with *p*-values < 0.05 considered to be significant.

### 2.5. Phylogenetic Analysis

A phylogenetic tree of *GDF9* gene coding sequence in 10 sheep breeds and other five species (human, goat, Norway rat, chicken, and pig) was constructed by the maximum likelihood method (model: K2P + G4) with IQ-TREE software [22]. For Cele Black sheep, Hu sheep, Small Tail Han sheep, Prairie Tibetan sheep, Valley Tibetan sheep, Bayinbuluke sheep, Tan sheep, and Wuzhumuqin sheep, the *GDF9* coding region sequences were extracted from the bam file of our previous whole-genome sequencing data with Samtools [23]. For Texel sheep, human, goat, Norway rat, chicken, and pig, sequences were obtained according to the NCBI (https://www.ncbi.nlm.nih.gov/, accessed on 27 May 2020) Reference Sequence (NC_019462.2, NC_000005.10, NC_030814.1, NC_005109.4, NC_006100.5, NC_010444.4, respectively).

### 2.6. Protein Structure Prediction

The secondary structure of GDF9 with and without missense mutants was analyzed using SOPMA [24]. Prediction of the tertiary structure of GDF9 with and without missense mutants was performed using Swiss-Model [25] with homology modeling. The GDF9 mature protein region was predicted using uniprot [26].

## 3. Results

### 3.1. Polymorphism Analysis of GDF9 Gene in Luzhong Sheep 

To determine the polymorphism of *GDF9* gene in Luzhong sheep with different litter sizes, the entire coding sequence of *GDF9* gene in 154 ewes were sequenced. By comparing their coding sequence with the known sequence of *GDF9* gene in Rambouillet sheep (GenBank NC_040256.1), eight mutation loci were identified in the *GDF9* gene, of which one mutation (g.41769606 T > G) was novel. The *GDF9* gene coding sequence in Luzhong mutton sheep was shown in Appendix A. The mutation information is summarized in Table 1 and the sequencing profiles around eight mutation loci are shown in Figure 1. The combined results from sequence data and alignment analysis of *GDF9* revealed five SNPs in intron 1 and three SNPs in exon 2 (Table 2). No mutation was detected in exon 1 of the *GDF9* gene coding sequence. These mutations were different from the *FecG^H^* mutation in African Rahmani sheep. By observing the genotypes of different mutation sites together, it was found that linkage exists between mutation loci g.41769246 A > G and g.41769002 A > G, g.41768501 A > G and g.41768485 G > A, and g.41769567 T > C and g.41769223 C > G, respectively. Of them, mutation g.41768485 G > A in exon 2 altered the amino acid (Val (V)–Ile (I)) at residue 332 and other two mutations in exon 2 were synonymous mutations (Table 2).

### 3.2. Population Genetic Analysis of SNPs in GDF9 and FecB

Population genetics analyses of eight SNPs in *GDF9* and *FecB* were performed. Results showed that g.41769246 A > G, g.41769002 A > G, and *FecB* mutations had moderate polymorphisms (0.25 < *PIC* < 0.5), whereas six other loci (g.41769606 T > G, g.41769574 G > A, g.41769567 T > C, g.41769223 C > G, g.41768501 A > G, and g.41768485 G > A) had low polymorphisms (*PIC* < 0.25) in Luzhong mutton sheep (Table 3). In addition, the Chi-square test revealed that eight loci of *GDF9* were under the Hardy–Weinberg equilibrium (*P* > 0.05), except *FecB* (*P* < 0.05) in Luzhong mutton sheep.

### 3.3. Association Analysis between Eight Loci in GDF9 and FecB with Litter Size in Luzhong Sheep

The results indicated that the g.41768501A > G and g.41768485 G > A loci were significantly associated with litter size (Table 4). For the two loci, the litter size of ewes with heterozygous genotype was the highest and significantly higher than that of ewes with the homozygous genotype mutation (*p* < 0.05). For the *FecB* mutation, the litter sizes of ewes with the mutant genotypes were significantly higher than that of ewes with wild genotype (*p* < 0.05, Table 4). Litter size of Luzhong sheep was significantly influenced by parity (*p* = 0.022). Additionally, there was no significant interaction effect between *FecB* and the two loci associated with litter size (both *p* = 0.057).

### 3.4. Construction of Phylogenetic Tree of GDF9 Gene for 10 Sheep Breeds and Other Five Animal Species

A phylogenetic tree of *GDF9* gene coding sequence for ten sheep breeds and five other animal species was constructed using the neighbor-joining (NJ) method, showing that four prolific sheep breeds (Cele Black, Small Tail Han, Hu, Luzhong) were most closely related (Figure 2). The sequences of 10 sheep breeds and goat are clustered together, and the sequences of the other species showed relatively large differences between them.

### 3.5. Prediction of the Protein Structure

For one missense mutation found in *GDF9* of sheep, the secondary and tertiary protein structures before and after the mutation at g.41768485 G > A were predicted, respectively. The results showed that secondary structure did not change significantly after the mutation (Figure 3A,B). Although the overall spatial structure of the protein did not change, there were obvious differences in the structure of the region containing V332I amino-acid substitution after the mutation (Figure 3C,D). Moreover, the locus is located in the mature protein region (Figure 3E), which may cause some changes in the function of this secreted factor in follicles.

## 4. Discussion

Several major genes affecting sheep prolificacy were identified, including *BMPR1B (FecB)* [13,14,15,16], *BMP15 (FecX)* [27,28,29], *GDF9 (FecG)* [2,4,30], *B4GALNT2 (FecL)* [31,32], and *Woodlands* [33]. *BMPR1B* has an important influence on follicular development and maturation, and its mutation A746G, namely *FecB*, can change primary follicle development and ovulation in sheep [10], mice [34], and goats [35]. BMP15 and GDF9 are the ligands of BMPR1B. *FecB* may lead to a weakened response of ewe granulosa cells to BMP15 and GDF9, which may lead to a lower level of granulosa cell proliferation in smaller follicles, and form an earlier LH (luteinizing hormone) response ability (the granulosa cells of the first mature follicle increase the production of steroids and thus inhibit the pituitary gland’s secretion of FSH (follicle-stimulating hormone), with only the follicles that express the LH receptor continuing to develop). Because more follicles have the ability to synthesize LH receptors, even if the FSH concentration drops, more follicles continue to develop during the preovulation period, eventually resulting in the ewe being able to release more eggs in an estrus period [36]. Furthermore, BMP4 and GDF5 can be used as natural ligands of BMPR1B, and the *FecB* mutation weakens the inhibitory effect of its ligand on steroid production of granulocytes, therefore, granulocytes can further differentiate and promote follicle maturation [37]. The attenuation of the *BMPR1B* signal caused by *FecB* mutation leads to an increase in the density of FSHR and LHR in granulosa cells, which can reduce apoptosis of the granulosa cells and increase the ovulation rate [10]. Moreover, changes in amino acid metabolism affected by the *FecB* genotypes relating to different rates of protein biosynthesis might affect the growth of the developing oocyte in follicular fluid [38]. The *FecB* mutation was detected to be significantly related to litter size in Hu sheep and Small-Tail Han sheep [13,39,40]. The present study found that the litter sizes of ewes with the mutation genotypes were significantly higher than those of ewes with the wild genotype. Therefore, *FecB* mutation could also be used as a molecular marker to improve litter size in Luzhong sheep breeding.

*GDF9* plays an important role in controlling ovarian physiology and enhancing oocyte developmental competence [41]. The *GDF9* mutation can affect the fertility and sterility by producing different effects on number of follicles and oocyte in ewes [1]. Previous studies showed that *GDF9* mutants were highly associated with litter size in sheep, mainly including *FecG^H^*, *FecG^T^*, *FecG^E^*, *FecG^F^*, and *FecG^V^* [30,42,43,44]. Among them, *FecG^H^*, *FecG^T^*, and *FecG^V^* mutations increase the ovulation rate and *FecG^E^* and *FecG^F^* mutations have additive effects on ovulation and litter size. In some cases, ewes with the homozygous mutant of *FecG^T^* and *FecG^V^* are infertile [30,43]. The *FecG^V^* can increase ovulation rate or litter size in heterozygote ewes and lead to infertility due to ovarian and uterine dysplasia in homozygote ewes. This infertility phenotype may be due to disruption of the PCSK (proprotein convertase subtilisin kexin) cleavage site of the GDF9 proprotein, preventing the conversion of GDF9 to dimeric, mature GDF9, which is the biologically active form of the protein [30]. For *FecG^T^*, incomplete follicle development leads to infertility in homozygous ewes, despite apparently normal oocyte activation and expression of some oocyte-specific genes (including those involved in ZP (zona pellucida) formation) [43]. In addition, a new mutation *FecG^A^* was detected in Araucana creole sheep, but this mutation was not significantly associated with litter size [8]. Another new mutation in exon 1 of the *GDF9* gene was found in Egyptian sheep, which is related to litter size [45]. Previous studies indicated that mutations in the coding region of the *GDF9* gene mainly exist in European sheep breeds [4,30,42], American sheep breeds [46], and a few Chinese indigenous sheep breeds. In the present study, eight mutations were detected in the *GDF9* gene of Luzhong sheep, which were different from the *FecG^H^* mutation found in African Rahmani sheep. Of them, one mutation (g.41769606 T > G) was novel and two tight linkage SNPs (g.41768501 A > G and g.41768485 G > A) were associated with litter size. The two loci g.41768501 A > G and g.41768485 G > A are similar to *FecG^T^* and *FecG^V^*, and the litter sizes of ewes with the heterozygous genotype are increased compared to homozygous individuals with mutated alleles. The mutations are located in the mature protein region of GDF9, which may affect the function and activity of the protein.

In order to further explore the changes in the structure of the protein after the mutation of g.41768485 G > A, the secondary structure and tertiary structure of the protein with and without the mutation were predicted, respectively. It was found that the region, including the mutation site, were always irregularly coiled before and after the mutation. However, there were some subtle differences in the three-dimensional structure. The slight changes in the three-dimensional structure may affect the binding affinity of GDF9 with its receptors, and further affect cumulus expansion and ovulation, which may have a final effect on ovulation rate or litter size in sheep; this needs further experimentation for verification.

*BMP15*, *GDF9,* and *BMPR1B* are located in the TGF-β pathway at the same time and can regulate cellular differentiation, follicular atresia, and oocyte maturation [47]. Specifically, BMP15 and GDF9 are secreted from oocytes and bind to the BMPR1B receptor on granulosa cells. So, it is possible that *BMP15*, *GDF9,* and *BMPR1B* have a synergistic effect on litter size in sheep. Demars et al. [27] found that BMP15 and GDF9 can form heterodimers, and the TGF-β pathway is inhibited after the two genes are mutated at the same time, eventually leading to an increase in ovulation as well. Moreover, GDF9 and BMP15 together can facilitate anti-Müllerian hormone expression, which is crucial for ovarian function [48]. According to previous studies, the ewes carrying *FecB* and *BMP15* mutants simultaneously showed higher litter sizes than individuals carrying the single mutant [13]. Hanrahan et al. [42] found that the ewes with mutants in both *GDF9* and *BMP15* had higher ovulation rates than those with any one mutation. However, little is known about the interactive effects of *GDF9* and *BMPR1B* genes. It was speculated that GDF9 cannot bind to the BMPR1B receptor, while BMP15 likely binds to the BMPR1B receptor to transmit a signal through phosphorylation of SMAD2/3 [49], which may be consistent with the results regarding the absence of obvious interactive effect of *GDF9* and *BMPR1B* genes on litter size in Luzhong mutton sheep.

Analysis of phylogenetic tree indicated that the genetic relationship of prolific sheep breeds is the closest and that they are clustered together. The reason for this is that their sequences are highly consistent, however, monotocous sheep breeds have one or more mutations in different loci compared with them. In this study, the evolutionary relationship of the *GDF9* gene sequence among different species was similar to the previous results of Monestier et al. [50] and Ahmadi et al. [51].

## 5. Conclusions

The present study identified eight SNPs, including one novel mutation in the *GDF9* gene of Luzhong sheep. Of them, g.41768501A > G and g.41768485 G > A were significantly associated with litter size in Luzhong sheep. The results preliminarily demonstrated that *GDF9* is a major gene affecting the fecundity of Luzhong mutton sheep and the above two loci might be potential genetic markers for improving litter size.

## Figures and Tables

**Figure 1 animals-11-00571-f001:**
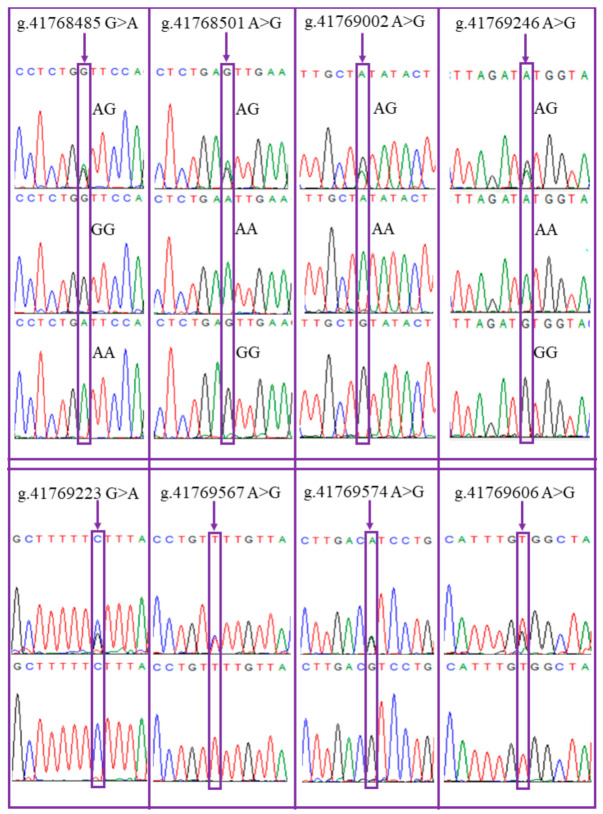
Sequencing profiles around eight mutation loci in the *GDF9* gene of Luzhong sheep.

**Figure 2 animals-11-00571-f002:**
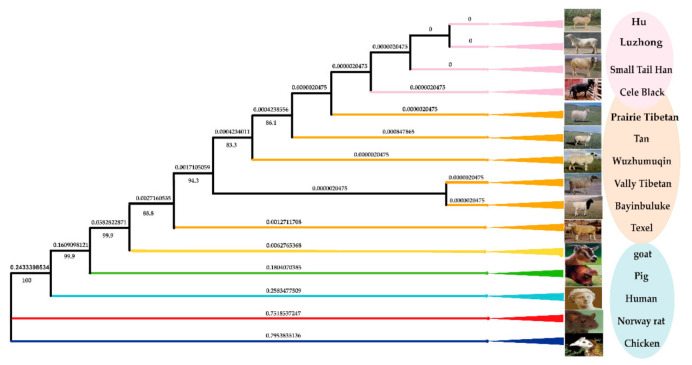
The phylogenetic tree of the *GDF9* gene in different species. The upper values refer to branch lengths values and the lower values refer to the bootstrap values measured in the number of substitutions per site. Note: The pink circle refers to the four prolific sheep breeds (Cele Black, Small Tail Han, Hu, Luzhong). The orange circle refers to mono-lamb sheep breeds (Prairie Tibetan, Tan, Wuzhumuqin, Valley Tibetan, Bayinbuluke, Texel). The light blue circle refers to different species.

**Figure 3 animals-11-00571-f003:**
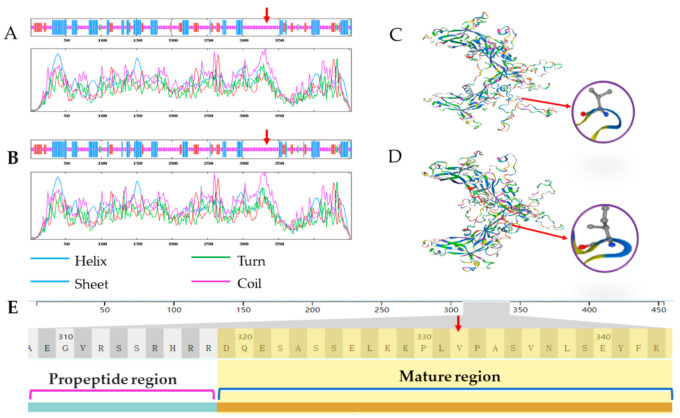
Secondary protein structure and tertiary protein structure for the GDF9 product before and after the mutation at g.41768485 G > A based on its amino acid sequence. (**A**) Secondary protein structure before the mutation; (**B**) Secondary protein structure after the mutation; (**C**) Tertiary protein structure before the mutation; (**D**) Tertiary protein structure after the mutation. (**E**) The mutation is located in the GDF9 mature protein region. Note: The red arrow refers to the g.41768485 G > A locus.

**Table 1 animals-11-00571-t001:** Primers used for the amplification of the *GDF9* gene.

Primer Name	Primer Sequence (5′–3′)	Annealing Temperature/°C	Amplified Fragment/bp
*GDF9*-F1	ACTGAATGAATAGGGTGTTG	58	1553
*GDF9*-R1	ATCTGTACCATATCTAAGTCC	57	
*GDF9*-F2	GTTTTGTTACAGTGGGTTTAGAGC	63	1507
*GDF9*-R2	GGCGCGGCATTTACACTGG	58	
*GDF9*-F3	TCAGCTGAAGTGGGACAA	60	694
*GDF9*-R3	ACACAGAAAATTTATGCCACTCAC	62	

**Table 2 animals-11-00571-t002:** The information of eight mutation loci in the *GDF9* gene of Luzhong sheep.

Region	Genomic Location (Chr5: Oar_v4.0/Oar_v3.1)	Wild-Type	Mutant	Mutation in Ensembl Database	AA Coding Residue	Amino Acid Change
Intron 1	41769606/41842523	T	G	No	-	-
	41769574/41842491	G	A	Yes	-	-
	41769567/41842484	T	C	Yes	-	-
	41769246/41842163	A	G	Yes	-	-
	41769223/41842140	C	G	Yes	-	-
Exon 2	41769002/41841919	A	G	Yes	159	Unchanged Leu (L)
	41768501/41841418	A	G	Yes	326	Unchanged Glu (E)
	41768485/41841402	G	A	Yes	332	Val (V)–Ile (I)

**Table 3 animals-11-00571-t003:** Population genetic analysis of eight SNPs for *GDF9* and *FecB* mutations in Luzhong mutton sheep.

Gene	Locus	Genotype Frequency			Allele Frequency		*PIC*	*He*	*Ne*	Chi-Square Test (*p*-Value)
*GDF9*	g.41769606 T > G	TT	GT	GG	T	G	0.013	0.013	1.013	0.935
		0.987	0.013	0.000	0.994	0.007				
	g.41769574 G > A	GG	AG	AA	G	A	0.013	0.013	1.013	0.935
		0.987	0.013	0.000	0.994	0.007				
	g.41769567 T > C	TT	CT	CC	T	C	0.083	0.087	1.095	0.555
		0.909	0.091	0.000	0.955	0.046				
	g.41769246 A > G	AA	AG	GG	A	G	0.325	0.408	1.690	0.821
		0.078	0.416	0.506	0.286	0.714				
	g.41769223 C > G	CC	GC	GG	C	G	0.083	0.087	1.095	0.555
		0.909	0.091	0.000	0.955	0.046				
	g.41769002 A > G	AA	AG	GG	A	G	0.325	0.408	1.690	0.821
		0.078	0.416	0.506	0.286	0.714				
	g.41768501 A > G	AA	AG	GG	A	G	0.229	0.263	1.357	0.874
		0.714	0.260	0.026	0.844	0.156				
	g.41768485 G > A	GG	AG	AA	G	A	0.229	0.263	1.357	0.874
		0.714	0.260	0.026	0.844	0.156				
*FecB*	g.29315643 A > G	AA	AG	GG	A	G	0.340	0.434	1.766	0.005
		0.416	0.532	0.052	0.682	0.318				

**Table 4 animals-11-00571-t004:** Least-square means and standard errors of litter size for different genotypes of nine loci in Luzhong ewes. Note: Different letters (^a^, ^b^) for the groups indicate differences (*p* < 0.05).

Gene	Locus	Genotype	Litter Size (Mean ± SD)
*GDF9*	g.41769606 T > G	TT	1.612 ^a^ ± 0.680
		GT	2.000 ^a^ ± 0.000
	g.41769574 G > A	GG	1.612 ^a^ ± 0.680
		AG	2.000 ^a^ ± 0.000
	g.41769567 T > C	TT	1.614 ^a^ ± 0.684
		CT	1.642 ^a^ ± 0.621
	g.41769223 C > G	CC	1.614 ^a^ ± 0.684
		CG	1.642 ^a^ ± 0.621
	g.41769246 A > G	AA	1.750 ^a^ ± 0.608
		GG	1.603 ^a^ ± 0.688
		AG	1.609 ^a^ ± 0.679
	g.41769002 A > G	AA	1.750 ^a^ ± 0.608
		GG	1.603 ^a^ ± 0.688
		AG	1.609 ^a^ ± 0.679
	g.41768501 A > G	AA	1.564 ^a^ ± 0.656
		AG	1.825 ^a^ ± 0.708
		GG	1.000 ^b^ ± 0.000
	g.41768485 G > A	GG	1.564 ^a^ ± 0.656
		AG	1.825 ^a^ ± 0.708
		AA	1.000 ^b^ ± 0.000
*FecB*	g.29315643 A > G	AA	1.094 ^b^ ± 0.342
		AG	1.988 ^a^ ± 0.596
		GG	2.000 ^a^ ± 0.730

## Data Availability

The gene sequence data presented in this study are available in Appendix A.

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
