# Peer review of "Polymorphism Detection of GDF9 Gene and Its Association with Litter Size in Luzhong Mutton Sheep (Ovis aries)"

_animals, 2021, doi:10.3390/ani11020571_

Round 1

Reviewer 1 Report

The manuscript would benefit from including litter size data from the 154 ewes employed in the study. Did 150 lambs have twins? etc. Can we group the ewes based upon litter size for the naive reader to understand the spread in the data relative to the mutations. 

"Thank you very much for your good suggestion! Previous studies indicated that mutations in the coding region of the GDF9 gene mainly exist in European sheep breeds (Mullen et al., 2014; Souza et al., 2014; Hanrahan et al., 2004; Mullen et al., 2013) and few have been detected in Chinese indigenous sheep breeds. Here, under the guidance of your suggestion, we compared the loci associated with litter size in this study between other three prolific sheep breeds (Cele Black, Small Tail Han, Hu) and five Mono-lamb sheep breeds (Prairie Tibetan, Tan, Wuzhimuqin, Oula) in china, however no mutation was found. Therefore, this content has not been added to the manuscript finally. Thanks a lot again!"

The authors should add a statement to the manuscript, and they should also include US breeds. 

Author Response

Dear Reviewer:

We are very grateful to you for the helpful comments and suggestions on our manuscript! We have revised the manuscript carefully according to your suggestions and replied each point that you raised as completely as possible. Please see the following itemized reply. Thanks a lot again!

1、The manuscript would benefit from including litter size data from the 154 ewes employed in the study. Did 150 lambs have twins? etc. Can we group the ewes based upon litter size for the naive reader to understand the spread in the data relative to the mutations.

Response: Thanks a lot for your question and suggestion! In present study, 154 ewes had detailed records of the litter size for two parities. According to statistics of litter size, 88 ewes have twins for at least one parity. As described in the article, the association analysis indicated that g.41768501A > G and g.41768485 G > A loci were significantly associated with litter size.

2、"Thank you very much for your good suggestion! Previous studies indicated that mutations in the coding region of the GDF9 gene mainly exist in European sheep breeds (Mullen et al., 2014; Souza et al., 2014; Hanrahan et al., 2004; Mullen et al., 2013) and few have been detected in Chinese indigenous sheep breeds. Here, under the guidance of your suggestion, Therefore, this content has not been added to the manuscript finally. Thanks a lot again!" The authors should add a statement to the manuscript, and they should also include US breeds.

Response: Thank you very much for the helpful suggestion! Now the sentences “Previous studies indicated that mutations in the coding region of the GDF9 gene mainly exist in European sheep breeds (Mullen et al., 2014; Souza et al., 2014; Hanrahan et al., 2004; Mullen et al., 2013), American sheep breed (Heaton et al., 2017) and few Chinese indigenous sheep breeds.” had been added in Line 237 of the Discussion part. And the literature about American sheep breed has also been checked and added to our article. Thank you for your suggestion again!

Reviewer 2 Report

The new version of the manuscript present several improvements. In particular, the association analysis is more convincing now about the role of the mutations found in litter size.

I have only one major comment, and a few minor ones:

* The phylogenetic analysis needs to be more informative about the relationships that its showing, and their robustness. I am missing in figure 2 some measure of robustness in the tree nodes. The length of the branchs should account for genetic distance, and in its current state it might be misleading, since we do not actually know anything about these distances. Also on this figure, why including 4 outgroups? what do the colored circles represent? they seem to suggest that Luzhong is much closer to Hu than Prairie Tibetan, but again, are they? how much? The authors also mention on the results and discussion that Luzhong clusters closer to other prolific breeds. The discussion point is weak in its current state, and needs more justification and references.

Minor comments:

- Title: wonder if it should also mention FecB, and replace "the" by "its" (or "their" in FecB is included). There is also a trailing "1" that should be removed.

- line 81: wonder if providing identifiers that help researchers querying these loci would be better than providing the and URL.

- section 2.4: mathematical expressions might require citations (and other parameters and tests mentioned throughout the test, e.g. Duncan test in this section?), and would be better displayed one per line.

- line 104: RStudio is an IDE for R. R and its version number should be references instead, as well as the package (eg. glm) used and its version.

- line 117 and others: Provide references, instead of URLs, and version numbers for all software (eg. including SOPMA, MEGA etc)

- Discussion: I'd advocate to be more brief about the molecular functions under study, and elaborate more on the results obtained, and their potential implications.

Author Response

Dear Reviewer:

We are very grateful to you for the helpful comments and suggestions on our manuscript! We have revised the manuscript carefully according to your suggestions and replied each point that you raised as completely as possible. Please see the following itemized reply. Thanks a lot again!

The new version of the manuscript present several improvements. In particular, the association analysis is more convincing now about the role of the mutations found in litter size.

I have only one major comment, and a few minor ones:

1、* The phylogenetic analysis needs to be more informative about the relationships that its showing, and their robustness. I am missing in figure 2 some measure of robustness in the tree nodes. The length of the branchs should account for genetic distance, and in its current state it might be misleading, since we do not actually know anything about these distances. Also on this figure, why including 4 outgroups? what do the colored circles represent? they seem to suggest that Luzhong is much closer to Hu than Prairie Tibetan, but again, are they? how much? The authors also mention on the results and discussion that Luzhong clusters closer to other prolific breeds. The discussion point is weak in its current state, and needs more justification and references.

Responses: Thank you very much for pointing out the problem! At present, we have adopted a more reliable and accurate software (IQ-TREE) for phylogenetic tree construction. Therefore, the phylogenetic tree has been updated now, and the genetic distance and the bootstrap values have been clearly displayed in the Figure 2. Additionally, the notes for the different colored ellipses have been added to the legend. In addition to analyze the clustering relationship of the gene between sheep breeds, we also need to analyze the evolution of the gene sequencebetween species similar to these references (Monestier et al., 2014; Ahmadi et al., 2016). The sequences of the four breeds (Cele Black, Small Tail Han, Hu, Luzhong) are same exactly, therefore the genetic relationship between Luzhong and Hu sheep is closer than relationship between Luzhong and Prairie Tibetan, and their genetic distance between Luzhong and Hu sheep is 0 as shown in Figure 2. However, the sequence of Prairie Tibetan sheep is different from that of Hu sheep. Thank you for your suggestion again!

References

Monestier, O.; Servin, B.; Auclair, S.; Bourquard, T.; Poupon, A.; Pascal, G.; Fabre, S. Evolutionary origin of bone morphogenetic protein 15 and growth and differentiation factor 9 and differential selective pressure between mono- and polyovulating species. Biol Reprod 2014, 91, 83.

  1. Ahmadi, F.A., R. Talebi, A. Farahavar and S.M.F. Vahidi. Investigation of GDF9 and BMP15 polymorphisms in Mehraban sheep to find the missenses as impact on protein. Iranian Journal of Applied Animal Science 2016, 6, 863-872.

Minor comments:

2、Title: wonder if it should also mention FecB, and replace "the" by "its" (or "their" in FecB is included). There is also a trailing "1" that should be removed.

Responses: Thank you very much for your friendly suggestion! FecB gene is a known major gene that affects the litter size in Chinese sheep, however, there are few reports on the effect of GDF9 gene on litter size in Chinese sheep. The main new finding of this study is the influence of the GDF9 gene on litter size in Luzhong sheep. Therefore, the title has not changed after consideration.

3、line 81: wonder if providing identifiers that help researchers querying these loci would be better than providing the and URL.

Responses: Thank you very much for your suggestion! Currently, for all eight SNPs, the position of them in version 3.1 and 4.0 has been shown in Table 2. This URL is just to help readers to check other relevant information of these loci.

4、section 2.4: mathematical expressions might require citations (and other parameters and tests mentioned throughout the test, e.g. Duncan test in this section?), and would be better displayed one per line.

Responses: Thank you very much for your suggestion! We have added references for mathematical expressions, and displayed them on one per line. Please see the revised manuscript.

5、line 104: RStudio is an IDE for R. R and its version number should be references instead, as well as the package (eg. glm) used and its version.

Responses: Thank you very much for your suggestion! Now we have updated this sentence in Line 108 and added corresponding reference and software package.

6、line 117 and others: Provide references, instead of URLs, and version numbers for all software (eg. including SOPMA, MEGA etc)

Responses: Thank you very much for your suggestion! We have added references for  the software. Please see the revised manuscript, thanks!

7、Discussion: I'd advocate to be more brief about the molecular functions under study, and elaborate more on the results obtained, and their potential implications.

Responses: Thanks a lot for your suggestion! However, another reviewer mentioned that the molecular functions of each candidate gene are very important and need to be added in the discussion section, therefore, we retain the description of the molecular function in the discussion section. In additional, for the missense mutation, we found that the mutation is located in the mature domain of the protein and affects the tertiary structure of the protein, and their possible influence on ovulation has also been discussed. Finally, their potential implications as molecular markers had been also mentioned in this manuscript. Thanks again for your good suggestions!

Reviewer 3 Report

Dear Authors,

I see that the English was improved, but there are still some mistakes. English native speaker really must check the whole article after the revision and before publishing.

I have no further doubts to the presented mutations. The eight mutations presented in the revised manuscript look like normal polymorphic sites. Thank you for presenting improved chromatograms. However, please improve figure 1 – make it better quality and sharpen it.

I didn’t have access to the FASTA sequence results (they should have been in supplementary materials), so I couldn’t use it or check it.

The reference nr 38, placed in the References, was not cited in the text. I didn’t find in the text. Please remove it from the References or correct it in the article text.  
38. Mullen, M.P.; Hanrahan, J.P.; Howard, D.J.; Powell, R. Investigation of prolific sheep from UK and Ireland for 381 evidence on origin of the mutations in BMP15 (FecX(G), FecX(B)) and GDF9 (FecG(H)) in Belclare and Cambridge 382 sheep. PLoS ONE 2013, 8, e53172.

In general, the paper was improved and can be published after minor revision and English corrections.

Author Response

Dear Reviewer:

We are very grateful to you for the helpful comments and suggestions on our manuscript! We have revised the manuscript carefully according to your suggestions and replied each point that you raised as completely as possible. Please see the following itemized reply. Thanks a lot again!

Dear Authors,

1、I see that the English was improved, but there are still some mistakes. English native speaker really must check the whole article after the revision and before publishing.

Responses: Thank you very much for your suggestions! We carefully checked the spelling and grammar of the full text, and made changes.

2、I have no further doubts to the presented mutations. The eight mutations presented in the revised manuscript look like normal polymorphic sites. Thank you for presenting improved chromatograms. However, please improve figure 1 – make it better quality and sharpen it.

Responses: Thanks for your suggestion! Now we have improved Figure 1 as shown in below.

3、I didn’t have access to the FASTA sequence results (they should have been in supplementary materials), so I couldn’t use it or check it.

Responses: Thank you very much for the problem! The sequence has been already submitted to the system when we submitted the article last time. Therefore, we expect that this attachment should be available for download after the article is published.

4、The reference nr 38, placed in the References, was not cited in the text. I didn’t find in the text. Please remove it from the References or correct it in the article text.  
38. Mullen, M.P.; Hanrahan, J.P.; Howard, D.J.; Powell, R. Investigation of prolific sheep from UK and Ireland for 381 evidence on origin of the mutations in BMP15 (FecX(G), FecX(B)) and GDF9 (FecG(H)) in Belclare and Cambridge 382 sheep. PLoS ONE 2013, 8, e53172.

Responses: Thanks a lot for your reminder! We have checked the references carefully and modified them.

5、In general, the paper was improved and can be published after minor revision and English corrections.

Response: Thank you very much for your suggestion! We have revised the manuscript carefully according to your suggestions.

This manuscript is a resubmission of an earlier submission. The following is a list of the peer review reports and author responses from that submission.

Round 1

Reviewer 1 Report

Please revise the manuscript to the third person perspective. 

"our" in the summary, and please remove the colloquial statements such as "on the other hand", "In order to" (l74) , "in other words" (l161) "so far" l185 

The authors present a realatively simplistic study where they examined polymorphisms in GDF9 in the Luxhon sheep to better understand litter size in the animals. 

The authors did not compare the polymorphisms across breeds of sheep with different litter sizes, and much of the manuscript is modeling. 

the major deficiency of the study is that the authors do not compare their polymorphisms to any performance data from their flock

Can the authors make any speculations about the Finn/Dorset crosses employed in Doug Hogue's star system relative to their SNPs? 

Author Response

Dear Reviewer:

We are very grateful to you for the helpful comments and suggestions on our manuscript! We have revised the manuscript carefully according to your suggestions and replied each point that you raised as completely as possible. Please see the following itemized reply. Thanks a lot again!

1、Please revise the manuscript to the third person perspective.

Response: Thank you very much for your kind suggestion! We have checked the manuscript carefully and revised the manuscript to the third person perspective.

2、"our" in the summary, and please remove the colloquial statements such as "on the other hand", "In order to" (l74), "in other words" (l161) "so far" l185.

Response: Thank you very much for your suggestion! These contents had been revised according to your advice.

3、The authors present a realatively simplistic study where they examined polymorphisms in GDF9 in the Luxhon sheep to better understand litter size in the animals.

The authors did not compare the polymorphisms across breeds of sheep with different litter sizes, and much of the manuscript is modeling.

Response: Thank you very much for your good suggestion! Previous studies indicated that mutations in the coding region of the GDF9 gene mainly exist in European sheep breeds (Mullen et al., 2014; Souza et al., 2014; Hanrahan et al., 2004; Mullen et al., 2013) and few have been detected in Chinese indigenous sheep breeds. Here, under the guidance of your suggestion, we compared the loci associated with litter size in this study between other three prolific sheep breeds (Cele Black, Small Tail Han, Hu) and five Mono-lamb sheep breeds (Prairie Tibetan, Tan, Wuzhimuqin, Oula) in china, however no mutation was found. Therefore, this content has not been added to the manuscript finally. Thanks a lot again!

Reference:

Mullen, M.P.; Hanrahan, J.P. Direct evidence on the contribution of a missense mutation in GDF9 to variation in ovulation rate of Finnsheep. PLoS ONE 2014, 9, e95251.

Souza, C.J.; McNeilly, A.S.; Benavides, M.V.; Melo, E.O.; Moraes, J.C. Mutation in the protease cleavage site of GDF9 increases ovulation rate and litter size in heterozygous ewes and causes infertility in homozygous ewes. Anim Genet 2014, 45, 732-739.

Hanrahan, J.P.; Gregan, S.M.; Mulsant, P.; Mullen, M.; Davis, G.H.; Powell, R.; Galloway, S.M. Mutations in the genes for oocyte-derived growth factors GDF9 and BMP15 are associated with both increased ovulation rate and sterility in Cambridge and Belclare sheep (Ovis aries). Biol Reprod 2004, 70, 900-909.

Mullen, M.P.; Hanrahan, J.P.; Howard, D.J.; Powell, R. Investigation of prolific sheep from UK and Ireland for evidence on origin of the mutations in BMP15 (FecX(G), FecX(B)) and GDF9 (FecG(H)) in Belclare and Cambridge sheep. PLoS ONE 2013, 8, e53172.

4、The major deficiency of the study is that the authors do not compare their polymorphisms to any performance data from their flock.

Response: Thank you very much for your friendly suggestion! Since litter size is one of the most important economic traits which determine the economic benefits of modern sheep farming, the litter size was paid more attention in present study. The litter size for both parities has been recorded seriously and the association between polymorphisms in GDF9 and litter size was analyzed in this study. The previous studies illustrated that the candidate gene GDF9 is a major gene for litter size. We tried to search other references but did not retrieve the literature indicating that this gene is related to other phenotypes such as weight, height and so on. So, we didn’t measure those phenotypes ago. We really appreciate your suggestions! In the further analysis of other genes, we will do as you suggest.

5、Can the authors make any speculations about the Finn/Dorset crosses employed in Doug Hogue's star system relative to their SNPs?

Response: Thank you very much for your suggestion! According to your reminder, we have checked the relevant literature and did not find studies on the SNP of GDF9 gene in Finn/Dorset crosses. However, we found a report that the mutation V371M of GDF9 gene (which increased ovulation rate) was present in Finnsheep (Mullen and Hanrahan, 2014). Therefore, it is likely that the mutation also existed in Finn/Dorset crosses employed in Doug Hogue's star system and the mutation could be used as a potentially molecular marker for increasing ovulation in Finn/Dorset sheep.

Reference:

Mullen MP, Hanrahan JP: Direct evidence on the contribution of a missense mutation in GDF9 to variation in ovulation rate of Finnsheep. PloS one 2014, 9(4): e95251.

6Other illustration:

Dear reviewer, we are truly grateful to all reviewers for your helpful suggestions for our manuscript! We carefully checked each polymorphic locus again, and found a little problem, which has been corrected now. Correspondingly, the manuscript has also been revised carefully. Details are as follows. The presence of polyA structure in intron led to the relatively high background peaks at 4 loci (g.41769788 C > T, g.41769731 T > A, g.41769715 C > T, g.41769599 C > T) in original sequencing profiles, thus it affects the determination of genotype. For this problem, we redesigned the primer avoiding the polyA area and the new sequencing results indicated that the 4 loci aren’t polymorphic. Meanwhile, the sequencing results confirmed that the polymorphism at other 8 loci is correct. After careful checking, we are sure that the current results in the revised manuscript are accurate. Additionally, another reviewer suggested to change the methods of statistical analysis. Therefore, in the revised manuscript, the association analysis between polymorphic loci of GDF9 gene, FecB and litter size was performed using a general linear model procedure. Thanks a lot for your helpful suggestions again!

Reviewer 2 Report

The authors present a study on potential SNPs associated with fecundity traits in sheep, an important topic in animal breeding. However, I find important issues with the methods and results, that need to be addressed to provide evidence that the associations found are genuine, and make this worth publishing

Most importantly, what is central in this article is the association of 12 candidate SNPs to fecundity. However, the statistical analysis used for this is a Chi-squared test. This statistic is completely missused here. A Chi-squared test is used to determine whether some observed frequencies are different from the expectation or not. How can this possibly be used to associate litter size with genotypes? it cannot. The authors should use an actual association analysis, for example regression, over individual sheep parity values, and associate these with individual genotypes. This also allows to control for interaction with FecB genotypes, between 1st and 2nd parities, aminoacide change, etc., as well as to avoid problems derived from low statistical power (eg. introducing just two summary values to contrast). Since this analysis is central to the article, their results should be consolidated. Eg, using a second type of analysis, or including a validation scheme (eg. with training/test sets). In its current state, I do not find any convincing evidence for any of the mutations reported.

Table 2. Are AA coding residue and change columns incomplete? if only one mutation produces aminoacide change, are differences in litter size expected to be really associated to all 12 mutations, or just this nonsynonimous mutation? how could the other, synonymous mutations contribute to change in litter size? this all needs to be properly discussed. In it current state, the importance of SNPs found seems overstated, unless much better evidence is provided.

The discussion on these findings is also very reduced compared to other, more secondary points.

Minor comments:

- The type of assocation analysis used is important. Include it in the Abstract.
- Are there previous results on GDF9 or FecB mutations in the breeds White header Dorper and Hu, from which the breed in study is derived? Are the mutations found in Luzhong, de novo, or did they already exist in the parental breeds?
- Since Luzhong is a mixed breed, to which extent heterosis could contribute to its fecundity? Do we know something about heterozigosity in this and its parental breeds, genome-wide or at the loci in question? I think this is worth discussing point.
- What is the coverage of the sequencing? what sequencing technology? this is important, how else can we know if some of these mutations are sequencing errors or not?
- how were mutations called?
- were individuals/sequences pre-processed is some way? eg. with MAF, Hardy-Weinberg, inbreeding criteria? why? why not?
- section 2.3. why the sequencing schedule is different from that of GDF9?
- lines 89-92. This needs rephrasing. It is not clear how these values were computed, nor how HW equilibrium is determined.
- line 115. Revise table, figures and references numbers. In general, Tables and figures require better head/foot text so that they are more self-contained
- Figure 1. it is hard to follow without proper labelling.
- line 146. This is the first instance of "parity" in the text, and something should be say about it in Materials and Methods.

Author Response

Dear Reviewer:

We are very grateful to you for the helpful comments and suggestions on our manuscript! We have revised the manuscript carefully according to your suggestions and replied each point that you raised as completely as possible. Please see the following itemized reply. Thanks a lot again!

Reviewer 3 Report

The manuscript by Wang F., Pan  L., Wang X., He X., Zhang R., Tao L., La Y., Ma L., Chu M. and Di R., entitled “Novel Mutations of GDF9 Gene and their Association with Litter Size in Luzhong Mutton Sheep (Ovis aries)”, describes GDF9 gene polymorphism and its association with litter size in Luzhong Mutton Sheep. The article’s topic is interesting, but not very novel - the GDF9 gene has been already well described in many scientific papers and the Authors focused only on one native breed Luzhong Sheep. I have some doubts according to four shown mutation (described below). However, the study broaden the knowledge about GDF9 influence on litter size in sheep and should be published.

The text is understandable. However English native speaker must check the whole article after Authors revision. There are some mistakes (grammar and spelling), for instance the word “spices” in lines 164, 165 – I am sure that the Authors meant “animal species” not “animal spices”, which do not pass to the content.

  1. Abstract and Simple Summary – well constructed.
  2. Introduction is short, but informative. It gives good background for the study.
  3. Remarks and questions to the Methods:
    • Did the Authors sequenced fragments with forward and reverse primer, or with only one of them? Were the primers used for the sequencing the same as shown in table 1? Add this information.
    • The link in lines 83-84 works, however it shows the variant table for GDF9 Chr5: Oar_v3.1. According to table 2 (page 4), the Authors showed the location of mutations looking into Chr5: Oar_v4.0. The mutation’s locations may be different in different variants of the genomic data. If the location of the mutations in Table 2 is according to Oar version 4.0, then how can readers be sure that the mutations found by the Authors are truly absent in the version 3.1?
    • Did the Authors submitted their sequences to the public database or repository? If yes, please place in the article accession numbers. If not, place at least one FASTA sequence in the supplementary materials with marked mutations. It would help the readers to find and recognize the mutation positions.

  4. Results are presented in 4 tables and 3 figures.
    • I really appreciate the sequencing profiles in figure 1. They gave me a look to the sequencing quality. But looking through those profiles I have some doubts to the presented mutations. Due to my experience, the sequencing background is too high to be sure that there are for sure polymorphic sites (while sequencing only from one strand) in:
      - 41769599CT – the pointed polymorphic site’s peak (T, red) looks like the result of electrophoresis slip from previous peak (or polymerase slip), same as all smaller peaks in that sequence;
      - 41769788CT - the pointed polymorphic site’s peak (T, red) looks like the result of electrophoresis slip from previous peak (or polymerase slip), same as all smaller peaks in that sequence, especially, that “T” is also visible in the 41769788CC;
      - 41769731AT - the pointed polymorphic site’s peak (A, green) looks like the result of electrophoresis slip from previous peak (or polymerase slip), same as all smaller peaks in that sequence, especially, that “A” is also visible in the 41769731TT, smaller, like all other background peaks;
      - 41769715CT- the pointed polymorphic site’s peak (T, red) looks like the result of electrophoresis slip from previous peak (or polymerase slip), same as all smaller peaks in that sequence, especially, that “T” is also visible in the 41769715CC, smaller, like all other background peaks.
      It is easy to dispel those doubts by presenting the results of the sequencing from the other DNA strand (reverse?) or by confirming the results with other methods. Did the authors confirmed the presence of the mutations by two-strand sequencing (with forward and reverse primer) or other methods? Please show me the sequencing results for those mutation from the other strand, give the information about sequencing from both strands in the methods or show other confirmation for these three mutations. Without the additional confirmation I cannot accept those sites as polymorphic (mutations).
    • Check the table 3 and 4 – the first line (g.41769788) in both tables has different font than in rest of the table’s text. On purpose?
    • Table 5 is unclear for me. The Authors mentioned in the text (lines 157-158) and in the table’s title, that results placed in that table are for four loci in GDF9 and for FecB, but I can’t see the FecB polymorphism in the table. Please check it, explain better or correct.

  5. Remarks to the discussion

The discussion is comprehensive and well written.

5.1 Line 197: GDF5 or GDF9?

  1. The conclusions are maybe too optimistic, especially that two of mutations mentioned are doubtful, but they are based on the results.

  2. The references are properly chosen .

In general, the paper broaden the knowledge about GDF9 influence on fecundity in sheep and can be published, but only after major revision and English corrections.

Author Response

Dear Reviewer:

We are very grateful to you for the helpful comments and suggestions on our manuscript! We have revised the manuscript carefully according to your suggestions and replied each point that you raised as completely as possible. Please see the following itemized reply. Thanks a lot again!

The manuscript by Wang F., Pan L., Wang X., He X., Zhang R., Tao L., La Y., Ma L., Chu M. and Di R., entitled “Novel Mutations of GDF9 Gene and their Association with Litter Size in Luzhong Mutton Sheep (Ovis aries)”, describes GDF9 gene polymorphism and its association with litter size in Luzhong Mutton Sheep. The article’s topic is interesting, but not very novel - the GDF9 gene has been already well described in many scientific papers and the Authors focused only on one native breed Luzhong Sheep. I have some doubts according to four shown mutation (described below). However, the study broaden the knowledge about GDF9 influence on litter size in sheep and should be published.

1、The text is understandable. However English native speaker must check the whole article after Authors revision. There are some mistakes (grammar and spelling), for instance the word “spices” in lines 164, 165 – I am sure that the Authors meant “animal species” not “animal spices”, which do not pass to the content.

Response: Thank you very much for your good suggestions! The word “animal spices” was indeed misspelled, and we have modified it in the manuscript. We also carefully checked the spelling and grammar of the full text, and made changes.

2、Abstract and Simple Summary – well constructed.

3、Introduction is short, but informative. It gives good background for the study.

4、Remarks and questions to the Methods:

4.1 Did the Authors sequenced fragments with forward and reverse primer, or with only one of them? Were the primers used for the sequencing the same as shown in table 1? Add this information.

Response: Thanks a lot for your question! We sequenced fragments with forward and reverse primer, and the primers used for the sequencing are the same as shown in table 1. We had added this information in Method part.

4.2 The link in lines 83-84 works, however it shows the variant table for GDF9 Chr5: Oar_v3.1. According to table 2 (page 4), the Authors showed the location of mutations looking into Chr5: Oar_v4.0. The mutation’s locations may be different in different variants of the genomic data. If the location of the mutations in Table 2 is according to Oar version 4.0, then how can readers be sure that the mutations found by the Authors are truly absent in the version 3.1?

Response: Thank you very much for pointing out the problem! Currently, Oar version 4.0 is used more frequently, but only version 3.1 is available in the ensembl database. Therefore, based on your suggestion, we have added the location information of each locus for version 3.1 in the Table 2.

4.3 Did the Authors submitted their sequences to the public database or repository? If yes, please place in the article accession numbers. If not, place at least one FASTA sequence in the supplementary materials with marked mutations. It would help the readers to find and recognize the mutation positions.

Response: Thank you very much for your suggestion! We have added the FASTA sequence to the attachment, and we showed the mutations in green.

5、Results are presented in 4 tables and 3 figures.

I really appreciate the sequencing profiles in figure 1. They gave me a look to the sequencing quality. But looking through those profiles I have some doubts to the presented mutations. Due to my experience, the sequencing background is too high to be sure that there are for sure polymorphic sites (while sequencing only from one strand) in:

 - 41769599CT – the pointed polymorphic site’s peak (T, red) looks like the result of electrophoresis slip from previous peak (or polymerase slip), same as all smaller peaks in that sequence;
- 41769788CT - the pointed polymorphic site’s peak (T, red) looks like the result of electrophoresis slip from previous peak (or polymerase slip), same as all smaller peaks in that sequence, especially, that “T” is also visible in the 41769788CC;
- 41769731AT - the pointed polymorphic site’s peak (A, green) looks like the result of electrophoresis slip from previous peak (or polymerase slip), same as all smaller peaks in that sequence, especially, that “A” is also visible in the 41769731TT, smaller, like all other background peaks;
- 41769715CT- the pointed polymorphic site’s peak (T, red) looks like the result of electrophoresis slip from previous peak (or polymerase slip), same as all smaller peaks in that sequence, especially, that “T” is also visible in the 41769715CC, smaller, like all other background peaks.
It is easy to dispel those doubts by presenting the results of the sequencing from the other DNA strand (reverse?) or by confirming the results with other methods. Did the authors confirmed the presence of the mutations by two-strand sequencing (with forward and reverse primer) or other methods? Please show me the sequencing results for those mutation from the other strand, give the information about sequencing from both strands in the methods or show other confirmation for these three mutations. Without the additional confirmation I cannot accept those sites as polymorphic (mutations).

Response: Thank you very much for your good suggestion! After reanalysis, we found that the presence of polyA structure (AAAAAAGAAAAAAAAA) in intron led to the relatively high background peaks at 4 loci (g.41769788 C > T, g.41769731 T > A, g.41769715 C > T, g.41769599 C > T) in original sequencing profiles, thus it affects the determination of genotype. For this problem, we redesigned the primer avoiding the polyA area and the new sequencing results indicated that the 4 loci aren’t polymorphic. Meanwhile, the sequencing results confirmed that the polymorphism result at other 8 loci is correct. After careful checking, we are sure that the current results in the revised manuscript are accurate. Thanks again for your suggestions!

The forward and reverse sequencing peak maps containing 8 loci are as follows. (A): The sequencing profiles of g.41768501 A > G and g.41768485 G > A locus. The above is the forward sequencing peak. And the following is the corresponding reverse peak. (B), (C), (D), (E), (F), (G): The sequencing profiles of g.41769246 A > G, g.41769002 A > G, g.41769223 C > G, g.41769574 G > A, g.41769567 T > C, g.41769606 T > G locus, respectively. The left is the forward sequencing peak. And the right is the corresponding reverse peak.

6、Check the table 3 and 4 – the first line (g.41769788) in both tables has different font than in rest of the table’s text. On purpose?

Response: Thanks a lot for your reminder! We have checked and modified them.

7、Table 5 is unclear for me. The Authors mentioned in the text (lines 157-158) and in the table’s title, that results placed in that table are for four loci in GDF9 and for FecB, but I can’t see the FecB polymorphism in the table. Please check it, explain better or correct.

Response: In the revised manuscript, according to the other reviewer’s suggestion, we switched to a new analysis method (fixed effect model) and also found that there is no synergistic effect between GDF9 and FecB, therefore we deleted Table 5 to streamline the article.

8、Remarks to the discussion

The discussion is comprehensive and well written.

8.1 Line 197: GDF5 or GDF9?

Response: Thanks a lot! We carefully examined this line, “BMP4 and GDF5 can be used as natural ligands of BMPR1B” is correct.

9、The conclusions are maybe too optimistic, especially that two of mutations mentioned are doubtful, but they are based on the results.

Response: Thank you for your suggestion! We have revised the description of the conclusion.

10、The references are properly chosen.

In general, the paper broaden the knowledge about GDF9 influence on fecundity in sheep and can be published, but only after major revision and English corrections.

Response: Thank you for your suggestion! We have revised the manuscript carefully according to your suggestions.
